# Antimicrobial Resistance in African Great Apes

**DOI:** 10.3390/antibiotics13121140

**Published:** 2024-11-27

**Authors:** Coch Tanguy Floyde Tanga, Patrice Makouloutou-Nzassi, Pierre Philippe Mbehang Nguema, Ariane Düx, Silas Lendzele Sevidzem, Jacques François Mavoungou, Fabian H. Leendertz, Rodrigue Mintsa-Nguema

**Affiliations:** 1Department of Biology and Animal Ecology, Research Institute for Tropical Ecology (IRET/CENAREST), Libreville BP 13354, Gabon; patmak741@gmail.com (P.M.-N.); rodriguemintsa@yahoo.fr (R.M.-N.); 2Ecole Doctorale des Grandes Ecoles de Libreville, Libreville BP 3989, Gabon; 3Helmholtz Institute for One Health, Helmholtz-Centre for Infection Research, Fleischmannstrasse 42, 17489 Greifswald, Germany; ariane.duex@helmholtz-hioh.de (A.D.);; 4Unit of Research in Health Ecology (URES/CIRMF), Franceville BP 769, Gabon; 5Laboratoire d’Ecologie des Maladies Transmissibles (LEMAT), Université Libreville Nord (ULN), Libreville BP 1177, Gabon; 6Université Internationale de Libreville, Libreville BP 20411, Gabon

**Keywords:** antibiotics, antimicrobial resistance (AMR), African great apes (AGAs)

## Abstract

Background/Objectives: Antibiotic-resistant bacteria pose a significant global public health threat that demands serious attention. The proliferation of antimicrobial resistance (AMR) is primarily attributed to the overuse of antibiotics in humans, livestock, and the agro-industry. However, it is worth noting that antibiotic-resistant genes (ARGs) can be found in all ecosystems, even in environments where antibiotics have never been utilized. African great apes (AGAs) are our closest living relatives and are known to be susceptible to many of the same pathogens (and other microorganisms) as humans. AGAs could therefore serve as sentinels for human-induced AMR spread into the environment. They can potentially also serve as reservoirs for AMR. AGAs inhabit a range of environments from remote areas with little anthropogenic impact, over habitats that are co-used by AGAs and humans, to captive settings with close human–animal contacts like zoos and sanctuaries. This provides opportunities to study AMR in relation to human interaction. This review examines the literature on AMR in AGAs, identifying knowledge gaps. Results: Of the 16 articles reviewed, 13 focused on wild AGAs in habitats with different degrees of human presence, 2 compared wild and captive apes, and 1 study tested captive apes alone. Ten studies included humans working with or living close to AGA habitats. Despite different methodologies, all studies detected AMR in AGAs. Resistance to beta-lactams was the most common (36%), followed by resistance to aminoglycosides (22%), tetracyclines (15%), fluoroquinolones (10%), sulphonamides (5%), trimethoprim (5%), macrolide (3%), phenicoles (2%) and fosfomycin (1%). Conclusions: While several studies suggest a correlation between increased human contact and higher AMR in AGAs, resistance was also found in relatively pristine habitats. While AGAs clearly encounter bacteria resistant to diverse antibiotics, significant gaps remain in understanding the underlying processes. Comparative studies using standardized methods across different sites would enhance our understanding of the origin and distribution of AMR in AGAs.

## 1. Introduction

Infections caused by antibiotic-resistant bacteria have become a global public health problem, leading to reduced effectiveness of antimicrobial therapy and increased severity of infections [1]. Antimicrobial resistance (AMR) is recognized by the World Health Organization (WHO) as one of the greatest threats to human health [2]. Strikingly, by 2050 infections linked to AMR are expected to kill 10 million people per year [3,4].

The spread of AMR is generally attributed to the excessive use of antibiotics to treat humans and animals, and their use in the agro-industry. While AMRs can naturally occur in bacteria as defense mechanisms against antimicrobial compounds present in other life forms and the environment [5,6,7], the overuse of antibiotics is considered the driver of the emergence and spread of AMR. It works as a strong selection pressure favoring resistant bacteria, which can then contaminate the environment via humans, animals, and animal products. AMR can spread in the environment, e.g., along watercourses. Globally, resistant bacteria can spread with their hosts through human and animal movements, including human-induced processes like the trade of animals and their products, and naturally occurring processes like bird migrations. In addition, antimicrobial resistance genes (ARGs) can spread between bacteria through horizontal gene transfer. Environmental disturbances caused for example by climate change and human encroachment into wildlife habitats, can further impact the spread of AMR [8].

In recent years, more attention has been paid to AMR in different ecosystems. Resistant bacteria or ARGs appear to be present in most ecosystems, and most animal species, including many non-human primates (NHPs) [9,10]. African great apes (AGAs) may be especially good indicators of the human-induced spread of AMR into animal habitats. Firstly, AGAs occur in countries where antimicrobials are often used without prescription, and counterfeit and substandard drugs are often sold [11]. AGAs are our closest living relatives. Due to their genetic proximity to humans, they are susceptible to a very similar range of pathogens as our own species [12]. Therefore, AGAs can serve as sentinels and sources for emerging infectious diseases [12]. The same reasoning can be applied to AMR: Many human-adapted bacteria can likely infect AGAs and vice versa, and horizontal gene transfer can occur between closely related bacteria infecting humans and apes. The efficient transmission of bacteria between AGAs and humans was shown by the fact that the gut microbiome of NHPs in captivity is more similar to their human keepers than to wild conspecifics [13].

AGAs naturally inhabit tropical rainforests and wooded savannas along the equator. While humans encroach heavily on their natural habitats through hunting, agriculture, deforestation, tourism, and research [14,15], their behavioral flexibility also allows AGAs to persist in agroforestry mosaics dominated by human activity. Human activities create varying levels of contact between humans and AGAs, ranging from undisturbed areas with minimal anthropogenic impact, through co-utilized environments, to captive settings with close human–animal interactions, such as zoos and sanctuaries [16]. This may affect transmission of AMR, directly from humans or indirectly via co-used environments.

This review seeks to summarize the current state of knowledge on antimicrobial resistance in AGAs and identify relevant gaps for future research.

## 2. Results and Discussion

### 2.1. The Impact of Human Presence on AMR in AGAs

In this section, we focus on varying levels of human presence in the habitats of AGAs and their impacts on AMR detection. Thirteen studies targeted wild AGAs in environments with low or no human presence (e.g., strictly protected areas), or co-utilized environments (e.g., protected areas shared with local communities or used for ecotourism). In two studies, wild and captive settings were compared, and one study focused exclusively on captive apes.

Ten studies specifically compared AMR between AGAs (here, gorillas and chimpanzees) and humans working in or living near these habitats.

While some level of AMR was detected in all habitats and all investigated species, in environments with minimal human presence, the detected AMR showed little or no direct link to the resistances found in humans [5,17,18]. For example, Mbehang Nguema et al. (2015A) evaluated multidrug resistance in Enterobacteriaceae among gorillas in Gabon’s Moukalaba-Doudou National Park and nearby humans [19]. They found high resistance levels in humans but low levels in gorillas, with no apparent overlap in AMR. This pattern is also supported by Benavides et al. (2012) in Gabon’s Lopé National Park, Albrechtova et al. (2014) in Côte d’Ivoire’s Taï National Park, and a second study by Mbehang Nguema et al., in 2021 across three Gabonese parks [5,7,20]. These studies suggest that the resistance found in AGAs in strictly protected areas may be intrinsic rather than anthropogenic. However, an exception was noted by Schaumburg et al. (2012A), who detected penicillin resistance in a chimpanzee from Taï National Park, possibly linked to treatment with benzylpenicillin during a respiratory disease outbreak [18].

In co-utilized environments, the overlap in AMR was more apparent. For example, Parsons et al. (2021) found that in Tanzania’s Mitumba and Kasekela national parks, wild chimpanzees from Mitumba showed higher levels of sulfonamide resistance potentially due to closer contact with densely populated human areas [11]. Similarly, Weiss et al. (2018) reported that the most frequent multidrug resistance profiles in *E. coli* from humans in Uganda’s Kibale and Bwindi National Parks were also prevalent in livestock and wild primates. This indicates the widespread presence of resistant bacteria across species in such shared environments [21]. Rwego et al. (2008) and Goldberg et al. (2007) both confirmed that overlapping environments influence the rates and transmission patterns of resistant bacteria between humans and AGAs [22,23].

When comparing wild and captive AGAs, studies by Campbell et al. (2020) in the Republic of Congo and Bager et al. (2022) in Uganda indicated that the host’s lifestyle (i.e., captive or wild) had a greater influence on gut microbiome composition than geographic proximity [24]. Captive AGAs in close contact with humans showed higher frequencies of ESBL-producing bacteria, suggesting potential anthropogenic transmission [25]. Notably, Baron et al. (2021) found evidence of cross-species bacterial transmission between chimpanzees and termites in a Senegalese reserve, with shared clones of hospital-associated *K. pneumoniae* plasmids, highlighting interspecies and environmental AMR spread even in indirect human contact settings [26]. The comparison of the two studies by Schaumburg et al. (2012A and 2012B), conducted with the same methodology, supports the idea that captivity could lead to higher levels of resistance in AGAs compared to AGAs living in the wild. Indeed, the study conducted on wild chimpanzees and monkeys in Côte d’Ivoire and Gabon describes a low level of resistance, with the exception of an isolate found in a chimpanzee from Taï National Park previously treated with benzylpenicillin [18]. On the other hand, the study on captive chimpanzees in Uganda reports a high proportion of typically human-associated strains of *Staphylococcus aureus* carrying various antibiotic resistances [27]. This study concludes by highlighting that the reintroduction of primates from sanctuaries into the wild presents a risk to wild populations due to the introduction of human-adapted pathogens [27].

### 2.2. AMR Target and Testing Strategies

The reviewed studies investigated AMR using a range of targeted or more global investigative methods (Table 1). Eleven of the studies conducted were focused on Enterobacteriaceae in general [7,28], with six studies targeting specific bacteria belonging to the Enterobacteriaceae family, such as *Escherichia coli* [5,21,22,23] and *Klebsiella pneumonia* [20,25,26]. Enterobacteriaceae are an important target for AMR studies given, on the one hand, their ubiquity in the intestinal tracts of both humans and animals and on the other hand, their clinical relevance in the AMR context [29,30]. For example, infections caused by extended-spectrum beta-lactamase (ESBL)-producing Enterobacteriaceae, e.g., carrying CTX-M enzymes, have emerged as a major public health problem [31]. CTX-M enzymes are capable of degrading a wide range of antibiotics, such as cephalosporins and penicillins, making the treatment of infections much more difficult and limiting the available therapeutic options [32]. Additionally, resistance genes carried by Enterobacteriaceae can be horizontally transferred between different bacterial species allowing them to spread efficiently in ecosystems [33,34].

Moreover, for studying AMR in endangered wildlife, Enterobacteriaceae are a prime target due to their abundance in fecal matter, which can be collected non-invasively [20] without disturbing the animals in their natural behavior. Fecal samples can be collected either from habituated AGAs, i.e., animals that are regularly observed and used to the presence of humans, or under the sleeping nests of unhabituated AGAs [25,35]. In contrast, other types of samples from wild AGAs are a lot more difficult to obtain, requiring e.g., specifically trained veterinarians to perform necropsies of naturally deceased AGAs, or anesthesia of wild AGAs for live sampling. The latter is not only risky for the veterinarians but also for the animals, and is considered unethical when carried out for research purposes alone [23,36].

Only Schaumburg et al. studied *Staphylococcus aureus* and used nasal swabs from deceased or anesthetized animals, food residues, and bushmeat samples [18,27]. From an AMR perspective, *S. aureus* is an extremely relevant bacterium. Especially, methicillin-resistant *S. aureus* (MRSA) is a major public health issue due to its resistance to beta-lactam antibiotics like methicillin, oxacillin, and penicillin [37]. It can cause a range of infections, from mild skin issues to severe, life-threatening conditions such as septicemia, and pneumonia. These infections are often hard to treat due to the limited availability of effective antibiotics, which increases the risk of complications and deaths especially in vulnerable populations [38]. Studies on AMR in *S. aureus* are valuable for understanding the mechanisms of resistance and transmission pathways between humans, wildlife, and the environment [18,27]. As highlighted above, AMR was common in *S. aureus* strains collected from sanctuary chimpanzees [27], but no methicillin resistance was detected in the study of wild primates, and only a single isolate from a habituated wild chimpanzee from Taï National Park was resistant to penicillin. This animal, however, had previously been treated with benzylpenicillin during a respiratory disease outbreak. In addition, no MRSA was detected in bushmeat. Taken together, this suggests that MRSA is generally not widespread in wild great apes [18].

Bushmeat has previously been used to study zoonotic pathogens at the human–wildlife interface [39]. It offers the advantage of access to various tissue samples from diverse animals [39,40]. However, it also requires a sound ethical framework (e.g., no payments of any kind should be made for sampling bushmeat to avoid encouraging illegal hunting) [41]. For AMR studies, it should be kept in mind that bushmeat could be contaminated with bacteria from humans and the environment during hunting and handling, which may make it difficult to distinguish the source of detected AMR [42].

Regarding the AMR-targeted approaches, we identified three main methods used by the authors to study AMR in great apes. In one publication, the authors exclusively used the disc-diffusion method [5]. This method is based on phenotypic AMR characterization by culturing bacterial families such as Enterobacteriaceae and performing antibiotic susceptibility testing using small discs containing different antibiotics that are placed on the agar and prevent bacterial growth if the bacteria are not resistant. In twelve publications, the authors used an approach based on culture and sensitivity testing through disc diffusion followed by identification of the bacterial species using MALDI-TOF MS (achieved through mass spectrometry by analyzing the proteins present in bacterial cells) and other biochemical or molecular identification methods for specific bacteria such as *E. coli* or *K. pneumonia*. In some studies, a final molecular characterization through PCR and subsequent sequencing of PCR products to identify target gene sequences was performed [17,19,20,21,22,25,27]. Although this second approach characterizes various bacteria relevant to public and environmental health, it remains focused on a limited number of resistance genes and does not cover the entire AMR spectrum of the sample. Parsons et al. (2021) also used this targeted method for AMR characterization without first applying classical bacteriological culture and disc-diffusion methodology, opting directly for molecular analysis via PCR [11].

Global studies of entire resistomes are increasingly used to characterize the full range of resistance genes within a microbiome using high throughput sequencing (HTS) methods. Although Bager S.L. et al. (2022) and Baron et al. (2021) used a global approach by applying unbiased HTS to identify all ARGs (antibiotic resistance genes) present in their samples, their method remains non-exhaustive due to the preselection of bacteria using the culture method and disc-diffusion [25,26]. However, it is now known that ARGs are present in all ecosystems, including the environment and the human gut, in both pathogenic and non-pathogenic bacteria. The report by George et al. (2021) explored the entire resistome of the targeted microbiome by unbiased shotgun sequencing of fecal samples, examining all ARGs in pathogenic and non-pathogenic bacteria of this ecobiome [43]. However, this method also has limitations. Firstly, it requires very deep sequencing to describe the entire resistome, which makes up only a tiny fraction of the total DNA in the fecal sample. Secondly, it is complicated to link the detected ARG to a bacterial species that harbours the gene. This makes it complicated to assess pathogenic potential or clinical impact [44]. Thus, each approach to determining resistance has its strengths and limitations, and the use of one approach over another will depend on the research question being addressed such as identifying resistance genes, understanding their role in pathogenesis, or evaluating their clinical significance [45].

**Table 1 antibiotics-13-01140-t001:** Selected studies on AMR characterization in African great apes (AGAs) between 2002 and 2024.

Author and Year	Country	Species	AGA Tested	Location	Life Context	Sample Type	AMR Detection Method	Resistance Gene	Resistant Antibiotics	Associated Mechanism	Genetic Support
Janatova et al., 2014 [20]	Central African Republic	gorillas, Chimpanzee	63	Dzanga-Sangha, National Park	wild	faecal	Culture, disc diffusion, multiplex PCR	*qnrB33, qepA, strA, strB, sul1, sul2, blaCTX-M-15, blaCTX-M variant2, blaSHV-62, blaTEM-1, intI1(aadA1), intI1(dfrA12-orf-aadA2, intI1(dfrA7), tet(A), tet(B)*	vancomycin, teicoplanin, ampicillin, strep- tomycin, gentamicin, kanamycin, chlorampheni, tetracycline, erythromycin, ciprofloxacin,trimethoprim–sulfamethoxazole, linezolid, pristi-namycin and rifampicin, amoxycilin-clavulanicacid,ceftazidime, gentamicin.	Plasmid-transferase, Integron	plasmids
Parsons et al., 2021 [11]	Tanzania	Chimpanzee, baboons, humans and domestic animals	75	Gombe National Park	wild	faecal	multiplex PCR	*sul1; sul2; tetB*	tetracycline, streptomycin, trimethoprim-sulphonamide, sulphonamides compounds,	N	N
Baron et al., 2021 [26]	Senegal	Chimpanzee	48	Protected reserve	wild	faecal	Culture, disc diffusion, whole-genome sequencing of bacterial colonies	*blaOXA-1, blaSHV-28, blaTEM-1B, blaSHV-11, blaSHV-1-2a, blaTEM-1, blaCTX-M-15 (ESBL gene), blaOXA-48, blaKPC-2, aph(6)-Id, aph(3*′*)-Ib, aac(3)-IIa (GN), aac(6*′*)Ib-cr (AK), qnrS1, qnrB1, sul2, dfrA14, tet(A)*	Penicillinase, ESBL gene, Carbapenemase, Aminoglycoside(s), Fqa, Phenicol Sulphonamide(s), Tetracycline	N	N
Mbehang Nguema et al., 2015 [19]	Gabon	gorillas and humans	120	Moukalaba-Doudou National Park	wild	faecal	Culture, disc diffusion, biochemical identification, PCR/sequencing	*aadA, aadB, aph(3*′*), ampC, blaACT, blaSHV; cmrA, strA, strU; tetA; tetD; tetG; tetS; tetW*	betalactam; aminoglycoside; tetracycline; chloramphenicol	N	N
Mbehang Nguema et al., 2021 [7]	Gabon	Gorillas, other NHPs, other wildlife	125 for all species	National parks Moukalaba-Doudou, Loango and Lopé	wild	fecal	Culture, disk diffusion, double disk synergy test, VITEK, MALDI-TOFF	*N*	amoxicillin; amoxicillin + clavulanic, aztreonam; ceftazidime; cephalexin; chloramphenicol; cefotaxime; cefoxitin; levofloxacin; nalidixic acid; piperacillin; trimethoprim/sulfamethoxazole; temocillin; ticarcillin; ticarcillin + clavulanic acid; piperacillin + tazobactam;	N	N
Debora Weiss et al., 2018 [21]	Uganda	gorillas and other animals	N	Bwindi and Kibale Nationalpark	Wild	faecal	Culture, biochemical identification, disc diffusion, PCR/sequencing.	*dfrA1, aadA1*	ampicillin; chloramphenicol; doxycycline; tetracycline; streptomycin; sulfamethoxazole-trimethoprim; cephalothin;	dihydrofolate reductase, adenyltransferase	Cassette gene
Albrechtova et al., 2014 [17]	Ivory Coast	chimpanzee	43	Taı National Park	Wild	faecal	Culture, disc diffusion, double disk synergy test, multiplex PCR, MALDI-TOFF	*qnrB13, oqxA, blaCMY, blaACT, blaDHA, qnrB28*	amoxycilin-clavulanicacid, ciprofloxacin, ampicillin, Ceftazidime.	Plasmid-transferase (Conjugation)	plasmids
Schaumburg et al., 2012 [18]	Ivory Coast, Gabon	chimpanzee, gorillas, red end black colobus	31 and monkeys	Taï National Parc,	Wild	nasal swabs	Culture, biochemical identification, VITEK, PCR/sequencing.	*blaZ*	betalactamase	N	N
Frieder Schaumburg et al., 2012 [27]	Zambia, Uganda	chimpanzee	62	sanctuary	captive	nasal swabs	Culture, biochemical identification, VITEK, PCR/sequencing.	*BlaZ, mecA*	Penicillin methicillin	N	N
Mbehang Nguema, Okubo, et al., 2015 [28]	Gabon	gorillas	27	Moukalaba-Doudou National Park	Wild	faecal	Culture, disc diffusion, PCR/sequencing.	*tetB*	ampicillin, cefazolin, cefotaxime, streptomycin, tetracycline, ciprofloxacin, colistin, chloramphenicol and trimethoprim	N	N
Campbell et al., 2020 [24]	Republic of the Congo.	chimpanzee, gorillas and humans.	160 for all species	Nouabalé-Ndoki National Park	Wild and captive	fecal	functional metagenomics, shotgun metagenomic sequencing	*AAC(3)-VIIa, FmrO*	N	acetyltransferas, methylation of the ribosome	N
George et al., 2021 [43]	Nigeria,	chimpanzee	15	Boki Afi Wildlife Sanctuary	Wildlife Sanctuary	fecal	shotgun metagenomic sequencing	CfxA3, cfxA6, ANT(6)-Ia, aph(3’)-III, tet(32), tet(40),tet(O), tet(Q), tet(O/32/O), tetW, erm(B), erm(F)	Aminoglycoside, macrolide, Tetracycline, beta-Lactamase	adenyltransferases, phosphotransferases, ribosomal protection	plasmids
Rwego I.B. et al., 2008. [22]	Uganda	Humans, Mountain Gorillas, Livestock	66	Bwindi National Park	ecotourism, research, and wild	Rectal Swabs fecal	Culture, biochemical identification, disc-diffusion, PCR	*N*	Ampicillin, Cephalothin, Chloramphenicol, Doxycycline, Nalidixic acid, Streptomycin, Trimethoprim-sulfaxazole, and Tetracycline	N	N
Goldberg T.L. et al., 2006 [23]	Uganda	Humans chimpanzee	23	Kibale National Park	research tourism	Rectal Swabs fecal	Culture, biochemical identification, disc diffusion, PCR	*N*	ampicilline, cloramphenicol, ciprofloxacin, gentamicin, neomycin, sulfisoxazole; streptomycin, tetracycline, trimethoprim,	N	N
Benavides et al., 2012 [5]	Gabon	Humans, chimpanzee, other wildlife, lifestock	119	Lope National Park	Wild	Feces	Culture, disc diffusion	*N*	Ampicillin; Tetracycline; chloramphenicol; doxycycline; rifampin; streptomycin; and sulfamethoxazole.	N	N
Bager S.L., et al., 2022 [25]	Uganda	chimpanzee	86	Budongo Forest andNgamba Island	Wild and and captive	fecal	Culture, biochemical identification, disc diffusion, whole-genome sequencing of bacterial colonies	*blaCTX-M-15, blaTEM-1B, blaTEM-1C, blaSHV-11, blaSHV-12, blaOXA-1, blaLEN24, strA, strB, aadA1, aadA2, aadA5, aac(3)iia, aac(6’)ib-cr, oxqA, oxqB, qnrS1, fosA, dfrA12, dfrA17 mph(A), catA2, sul1, sul2*	Beta-blactams, aminoglycosides, fluoroquinolones, fosfomycins, macrolide-lincosamide-streptogramin, B, phenicols, sulfonamides cefpodoxime, ceftazidime, aztreonam, cefotaxime, ceftriaxone	N	N

Author and year: first author and year of publication; Country: country where the study was conducted; species: target species of the study; AGA tested: number of great ape samples tested location: study site; Life context: living environment of the target species; sample type: the samples used for the study; ARG detection method: laboratory methods for ARG detection; Resistance gene: the ARG detected during the analysis; resistant antibiotics: the antibiotics tested ineffective against the bacteria; Associated mechanism and genetics support detected; N: not described by authors.

Research on the frequency and distribution of AMR in wildlife in general, and AGAs in particular, is growing. This research ranges from purely phenotypic aspects of resistance to the intrinsic detection of genes conferring resistance. These resistance genes are classified into different families and superfamilies based on their mechanisms of action and structures [46]. These families and superfamilies of antibiotic-resistance genes can be present in a wide variety of pathogenic and commensal bacteria. Their diversity and distribution contribute to the spread of antibiotic resistance in bacterial populations and in the environment [47].

The most detected resistance among AGAs was resistance to beta-lactams, with a detection rate of 36% of targeted studies (Table 2). The most prevalent beta-lactamase genes were bla_CTX-M-15_ and bla_TEM-1_ from the *bla* gene superfamily of beta-lactams. These genes typically encode enzymes that can inactivate antibiotics [48,49,50] and were found in gorillas and chimpanzees from the Central African Republic and chimpanzees from Senegal and Uganda. Additionally, the study identified the *bla_ACT_* gene in gorillas from Gabon, chimpanzees from Côte d’Ivoire and Uganda, the *bla_SHV_* gene in gorillas from Gabon and chimpanzees from Uganda, and the *bla_Z_* gene in chimpanzees from Côte d’Ivoire. Finally, the *mecA* gene from the *AmpC* gene family, which is responsible for penicillin resistance through enzymatic inactivation, encoding a penicillin-binding protein (PBP2) acquired via lateral gene transfer [51], was found in chimpanzees from Zambia. Other beta-lactamase ARGs were detected only once in AGAs (Table 1).

**Table 2 antibiotics-13-01140-t002:** Percentage of ARGs by family in AGAs between 2002 and 2024.

Antibiotic Familiy	AMR Gene	Gene Detected by Antibiotic Family (%)	Host (African NHPs)	Ref.
Betalactams	*bla_CTXM-15_,*	36.2	Gorillas and Chimpanzee	[20,25,26]
	*bla_CTX-M_* *variant2*		Gorillas, Chimpanzee	[20]
	*bla_TEM-1_,*		Gorillas and Chimpanzee	[20,25,26]
	*bla_TEM-1B_*		Chimpanzee	[25,26]
	*bla_TEM-1C_*		Chimpanzee	[25]
	*bla_ACT_*		Gorillas, Chimpanzee	[17,19]
	*bla_SHV_*		Gorillas	[19]
	*bla_SHV-11_*		Chimpanzee	[25,26]
	*bla_SHV-12_*		Chimpanzee	[25]
	*bla_SHV-28_*		Gorillas, Chimpanzee	[26]
	*bla_SHV-62,_*		Chimpanzee	[20]
	*bla_LEN24_*		Chimpanzee	[25]
	*bla_SHV-1-2a_*		Chimpanzee	[26]
	*bla_OXA-1_*		Chimpanzee	[25,26]
	*bla_OXA-48_*		Chimpanzee	[26]
	*bla_KPC-2_*		Chimpanzee	[26]
	*bla_CMY_*		Chimpanzee	[17]
	*bla_DHA_*		Chimpanzee	[17]
	*bla_Z_*		Chimpanzee and Gorillas	[18,27]
	*CfxA3*		Chimpanzee	[43]
	*cfxA6*		Chimpanzee	[43]
	*AmpC*		Gorillas	[19]
	*mecA*		Chimpanzee,	[27]
Aminoside	*strA*	21.90	Gorillas, Chimpanzee	[20,25]
	*strB*		Gorillas, Chimpanzee	[20,25]
	*strU*		Gorillas	[20]
	*aadA*		Gorillas	[20]
	*aadA1*		Gorillas, Chimpanzee,	[20,25]
	*aadA2*		Gorillas, Chimpanzee	[20,25]
	*aadA5*		Chimpanzee	[25]
	*aadB*		Gorillas	[20]
	*ANT(6)-Ia*		Chimpanzee	[20]
	*aac(3)-IIa*		Chimpanzee	[25,26]
sulfamide	*AAC(3)-* *vIIa*	4.8	Chimpanzee	[26]
	*aac(6′)Ib-cr*		Chimpanzee	[25,26]
	*aph(3* *′* *)*		Gorillas	[19]
	*aph(3* *′* *)-III*		Chimpanzee	[26]
	*aph(6)-Id*		Chimpanzee	[20]
	*aph(3″)-Ib*		Chimpanzee	[20]
	*FmrO*		Gorillas, Chimpanzee	[20]
	*sul1*		Gorillas, Chimpanzee	[11,20]
	*sul2*		Gorillas, Chimpanzee	[11,20,26]
Trimethoprim	*dfrA7*	5.7	Gorillas, Chimpanzee	[20]
	*dfrA12*		Gorillas, Chimpanzee	[20,25]
	*dfrA14*		Gorillas, Chimpanzee	[21,26]
	*dfrA17*		Chimpanzee	[25]
fluoroquinolones	*qnrB1*	9.5	Gorillas, Chimpanzee	[20]
	*qnrB13*		Gorillas, Chimpanzee	[17]
	*qnrB28*		Gorillas, Chimpanzee	[17]
	*qnrB33*		Chimpanzee	[26]
	*qnrS1*		Chimpanzee	[25,26]
	*qepA*		Chimpanzee	[20]
	*oqxA*		Gorilllas, Chimpanzee	[17,25]
	*oxqB*		Chimpanzee	[25]
fosfomycin	*fosA*	1.0	Chimpanzee	[25]
Tetracycline	*tet(A)*	15.2	Gorillas, Chimpanzee	[19,20,26]
	*tet(B)*		Gorillas, Chimpanzee	[11,20,28]
	*tetD*		Gorillas	[19]
	*tetG*		Gorillas	[19]
	*tetS*		Gorillas	[19]
	*tet(32)*		Chimpanzee	[43]
	*tet(40)*		Chimpanzee	[43]
	*tet(O)*		Chimpanzee	[43]
	*tet(Q)*		Chimpanzee	[43]
	*tet(O/32/O)*		Chimpanzee	[43]
	*tetW*		Gorillas, Chimpanzee	[19,43]
phenicols	*cmrA*	2.0	Gorillas	[19]
	*catA2*		Chimpanzee	[25]
macrolide	*erm(B)*	2.9	Chimpanzee	[43]
	*erm(F)*		Chimpanzee	[43]
	*mph(A)*		Chimpanzee	[25]

Antibiotic familiy: the antibiotic classes for which the genes detected are resistant; AMR gene: the different resistance genes detected in our papers according to the families; % gene detection by antibiotic family: percentage of detection by gene families according to their presence/absence (cumulative genes per family according to their presence/absence and of all the genes present); Host (African Great Apes): Host species on which the study was carried out.

The second most prevalent resistances were those linked to aminoglycoside resistance (22%) and tetracycline (15%). Among aminoglycosides, the most common gene was *aadA1* (aminoglycoside nucleotidyltransferase), encoded by plasmids and integrons and belonging to the ANT superfamily (Streptomycin Nucleotidylylation) [52]. It was reported in gorillas and chimpanzees from the Central African Republic and Uganda, and in Ugandan gorillas (Figure 1). Several other genes encoding aminoglycoside resistance were detected in these studies (see Table 1), including the clinically significant *FmrO* gene, which confers a novel aminoglycoside resistance mechanism via methylation of the 16S ribosomal RNA [53], found in gorillas and chimpanzees from the Republic of the Congo. For tetracycline, *Tet(A)* and *Tet(B)* from the Major Facilitator Superfamily (MFS), associated with tetracycline inactivation by antibiotic efflux mechanisms [54], were found in gorillas and chimpanzees from the Central African Republic, gorillas from Gabon, and chimpanzees from Tanzania and Senegal.

### 2.3. Frequency and Distribution of AMR in African Great Apes

Resistance to fluoroquinolones (9.5%), trimethoprim (6%), and sulfonamides (5%) were the third, fourth, and fifth most prevalent in AGAs, respectively. Fluoroquinolone resistance is caused by genes that express various resistance mechanisms, such as the *qnr* family (*qnrB1*, *qnrB28*, *qnrB33*, and *qnrS1*), which encode plasmid-mediated quinolone-resistant proteins, and *qepA*, which encodes plasmid-mediated efflux pumps contributing to fluoroquinolone resistance [55]. These were found in the Central African Republic, Côte d’Ivoire, Uganda, and Senegal. Sulfonamide resistance encoded by *sul2* and *sul1* genes was found in gorillas and chimpanzees from the Central African Republic and in chimpanzees from Tanzania and Senegal. Trimethoprim resistance, on the other hand, is mediated by the *dfr* gene family (e.g., *dfrA7*, *dfrA12*, *dfrA14*, and *dfrA17*), which results in modifications of the target enzyme dihydrofolate reductase [56], detected in gorillas and chimpanzees from the Central African Republic, chimpanzees from Senegal, and gorillas from Uganda.

Chloramphenicol resistance (2.5%) found in Gabonese gorillas and macrolide resistance (2.5%) reported in Nigerian chimpanzees were the least prevalent in this study, with fosfomycin resistance (1%) reported in Ugandan chimpanzees.

### 2.4. Spread and Evolution of AMR

The transmission routes of AMR are complex. The authors of the analyzed papers have identified several transmission routes of ARGs and their evolution in different ecosystems. In some communities, unprotected wells located at the edge of forest habitats are used by both humans and livestock for water [21]. Interactions between humans, wildlife, and the environment can facilitate the transmission of bacteria and ARGs. Albrechtova et al. (2014) reported that the frequent need for antimicrobial treatment, coupled with poor sanitary conditions, creates ideal conditions for the widespread colonization of extended-spectrum beta-lactamases (ESBLs) in people and animals in rural communities [17,20]. This leads to the dissemination of plasmid-mediated resistance genes and the rapid spread of resistance in a given area. Great apes and other forest animals may encounter human food or feces, exposing them to the risk of contracting enteric bacteria and associated plasmid-mediated ARGs through the fecal–oral route. In sites where there is no livestock, the transmission of resistant bacteria from humans to wild non-human primates may occur unconsciously through tourists, air (wind, birds/bats), and wastewater, as already reported [19]. However, antibiotics are produced naturally in the environment by several soil bacteria; and AGAs microbiomes could thus acquire these ARGs through selection caused by exposure to naturally occurring antibiotics or through horizontal gene transfer from environmental bacteria allowing general maintenance of gene-plasmid associations [24]. The captive AGAs studied in the reviewed articles could be exposed to areas of human activity, through surfaces and food that humans had contact with. It is worth noting that the captive AGAs in the study conducted by Baron et al., (2021) were previously captured in the wild and brought to the Primate Research Institute (PRI), where they were initially kept in group cages and then transferred to individual cages [26]. This suggests that they could have been exposed to bacteria from the cage cleaners, which could potentially contain ARGs that may be transmitted from one primate to another in the same location. Also, programs to release non-human primates could be a factor in introducing AMR into the wild [57].

## 3. Materials and Methods

### 3.1. Literature Search Strategy

This review was conducted following the PRISMA protocol proposed by Page et al. in 2021 [58]. A web search was conducted across online data repositories (Google Scholar, Web of Science and Pubmed) to find relevant scientific articles related to AMR in AGAs. The keywords used for the online search were [AMR AND African Great Apes] OR [African great apes AND Antimicrobial Resistance Genes].

### 3.2. Inclusion and Exclusion Criteria

An article was considered eligible if (i) the publication date was between 2002 and 2024, (ii) the study was conducted in Africa, and (iii) the study included AGAs. The exclusion criteria were (i) articles that focused on topics other than AMR in AGAs, (ii) review articles, and (iii) articles published prior to 2002. We considered the year 2002 as the starting point for data collection because the majority of relevant records became available from this year onwards.

### 3.3. Data Extraction

Relevant information was extracted from the selected articles and used to create a comprehensive database containing author names, publication year, country of investigation, habitat, target animal, AMR detection method, number of individuals tested, whether they were wild or captive, resistance genes, different resistance phenotypes, resistance mechanisms, and genetic supports.

From the 200 articles identified through an online search using the designated keywords and based on the eligibility criteria, 16 relevant scientific publications were selected and used in this review (Figure 2).

## 4. Conclusions and Perspectives

Understanding the emergence of antimicrobial resistance (AMR) is a significant challenge in our time, as it is becoming increasingly widespread at various levels in the global environment. To effectively address this challenge, it is essential to gain a better understanding of the transmission, ecology, and evolution of AMR. Due to the genetic similarity between AGAs and humans, the bacterial pathogens and antimicrobial resistance determinants they carry can circulate in both directions. This study provides a comprehensive and up-to-date inventory of AMR identified in African great apes, noting the presence of resistance to beta-lactams, aminoglycosides, sulfonamides, trimethoprim, fluoroquinolones, fosfomycin, tetracyclines, phenicols, and macrolides. While several different techniques were applied and target bacteria were investigated, the reviewed studies show a trend that African great apes close to humans had higher levels of resistance compared to those living in the wild. Future studies should include a comprehensive analysis of antimicrobial resistance across great ape habitats. Such studies would allow for a more precise characterization of the presence or absence of ARGs, facilitate comparisons of resistance levels, document changes over time while considering different ecological factors (such as proximity to humans), and account for the origins and genetic variation in these genes. This would enable a better understanding of their evolution and transmission and, ultimately, contribute to the effective management of the growing problem of antimicrobial resistance in Africa and worldwide.

## Figures and Tables

**Figure 1 antibiotics-13-01140-f001:**
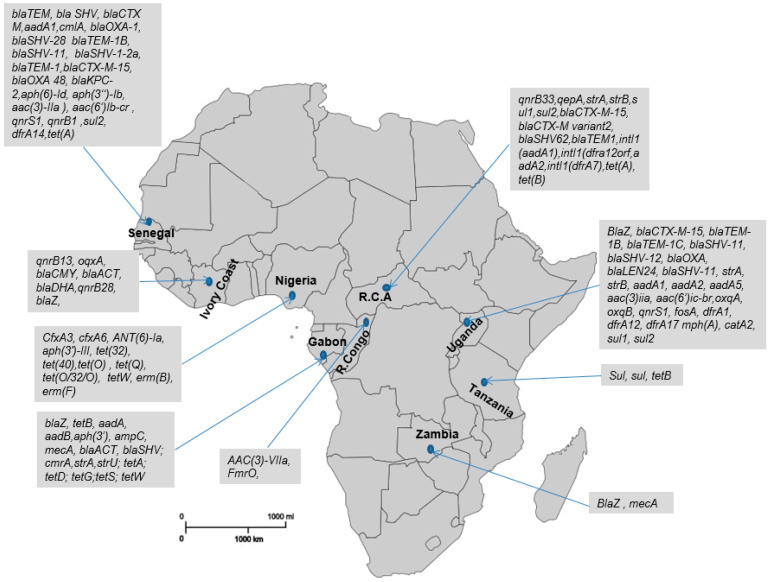
The distribution map of ARGs identified in AGAs between 2002 and 2024.

**Figure 2 antibiotics-13-01140-f002:**
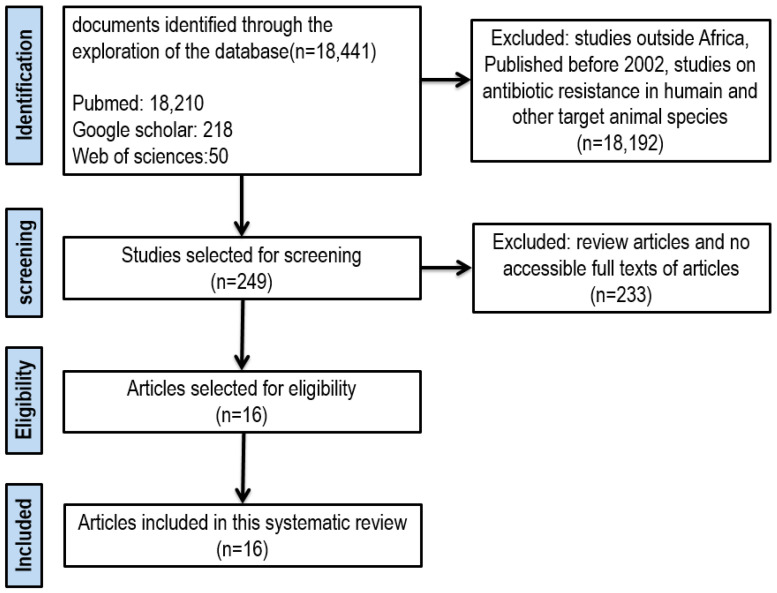
PRISMA flow chart for the search and selection of relevant papers for the systematic review.

## Data Availability

Data are contained within the article.

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
