# Peer review of "Antimicrobial Resistance in African Great Apes"

_antibiotics, 2024, doi:10.3390/antibiotics13121140_

Round 1
Reviewer 1 Report
Comments and Suggestions for Authors
The manuscript “An update on the Genetic characterization of Antibiotic Resistance in Non-Human Primates in Africa.” prepared by Coch Tanguy Floyde Tanga et al., reviewed the various antibiotic-resistant genes (ARGs) identified in African Non-Human Primates (NHPs) from 2002 to 2022. Given that antibiotic-resistant bacteria pose a significant threat to global public health and that ARGs can be found in all ecosystems, ARGs carried by NHP are critical for assessing antimicrobial resistance (AMR). Through the review of this article, we have a more comprehensive understanding of the current situation and possible causes of the resistance epidemic of β-lactam and other drugs. I am very interested in the review and analysis of this manuscript. However, there are still some questions about the manuscript that the author needs to clarify.
1 The data of "% detection gene" in Table 2 is puzzling.
2 The quality of Figure 2 is poor.
3 A comparison of drug resistance in other wild or farmed animals in Africa over the same period might be more beneficial to the theme.
4 Since it spans 20 years, a timeline comparison of the severity of resistance is necessary.
Author Response
Subject: Revisions
Dear reviewer of Antibiotics Journal,
Thank you for reviewing our work. We are writing this covering letter in accordance with the journal editor's request to explain, point by point, the details of the revisions made to the manuscript and our responses to your comments below. All revisions have been highlighted (in yellow) as suggested.
After receiving your initial comments a few months ago, we have reconsidered certain aspects of our manuscript. You will find in this new version some noticeable changes.
Initially, the scope was expanded to include non-human primates, but we have now narrowed it to focus on African great apes, as our research activities are solely centered on chimpanzees, gorillas, and bonobos. We have also broadened the subject to cover antimicrobial resistance as a whole, rather than focusing solely on resistance genes, which has resulted in an immediate change to the title of our manuscript. Accordingly, we have removed articles dealing with other primates and replaced them with other relevant studies, and we have extended the study period until 2024.
Regarding the sections of the article, we have removed the part dealing with genetic mechanisms and support, as recommended by one of the editors, and added two new sections that compare and discuss:
- Antimicrobial resistance (AMR) between great apes and humans, based on their level of overlap at each site.
- The methodologies used to analyze antimicrobial resistance and their targets.
Comments and Suggestions for Authors
The manuscript “An update on the Genetic characterization of Antibiotic Resistance in Non-Human Primates in Africa.” prepared by Coch Tanguy Floyde Tanga et al., reviewed the various antibiotic-resistant genes (ARGs) identified in African Non-Human Primates (NHPs) from 2002 to 2022. Given that antibiotic-resistant bacteria pose a significant threat to global public health and that ARGs can be found in all ecosystems, ARGs carried by NHP are critical for assessing antimicrobial resistance (AMR). Through the review of this article, we have a more comprehensive understanding of the current situation and possible causes of the resistance epidemic of β-lactam and other drugs. I am very interested in the review and analysis of this manuscript. However, there are still some questions about the manuscript that the author needs to clarify.
1-The data of "% detection gene" in Table 2 is puzzling.
Answers - The % genes detected data in Table 2 reflects the relative frequency (%) of different genes according to their presence/absence in each published paper. It shows us the genes that have been most abundantly detected over this period in African great apes, with reference articles for each gene.
2-The quality of Figure 2 is poor.
Answers- For figure 2, we have tried to represent the different genes detected in an Africa's map in order to have an idea of the distribution of these genes according to the countries and the papers reviewed. We will try to reproduce it with a better resolution.
3-A comparison of drug resistance in other wild or farmed animals in Africa over the same period might be more beneficial to the theme.
Answers- We completely agree with you on this point. However, our research group focuses on wild great apes as sentinels for emerging epidemics and the distribution of antimicrobial resistance. Since the available data for wild great apes is quite limited, we initially extended the study to include non-human primates. However, as mentioned earlier, we have revised our approach and decided to remain focused on great apes. We take note of this remark for a potential future study.
4-Since it spans 20 years, a timeline comparison of the severity of resistance is necessary.
Answers- Given the considerable lack of data on non-human primates, and therefore on great apes, as well as the specificity of the studies conducted (most studies are based on the specific characterization of resistance through PCR), it was impossible for us to make a chronological comparison without the risk of biasing the results. However, we attempted to make comparisons.
Best regards
Tanguy Tanga.
Reviewer 2 Report
Comments and Suggestions for Authors
In this review article, the authors evaluated the prevalence of antibiotic resistance genes in non-human primates over a 20 year period. The overall premise of the manuscript is of interest and the data presented are valuable. However, the manuscript needs to be shortened and revised.
1. The estimate is that 10 million people could die from AMR infections by 2050, not 2030, please correct this.
-
- O'Neill J Tackling drug-resistant infections globally: final report and recommendations. Review on Antimicrobial Resistance, London2016
-
- O'Neill J Antimicrobial resistance: tackling a crisis for the health and wealth of nations. Review on Antimicrobial Resistance, London2014
2. Table 1 requires significant attention and reformatting.
"antibiotic resilent" is that supposed to be "antibiotic resistance?"
"conjugaison" is that "conjugation"?
"gorilles" should be "gorillas"
gene names should be lower case
beta-lactamase genes should be written as follows blaSHV.
What does "associated mechanism" mean vs. "genetic support"? conjugation and plasmids are way to transfer resistance genes. What does "N" mean? nothing was reported?
There are drugs listed in the same sections and resistance mechanisms, and resistance mechanisms listed in the associated mechanism.
3. Table 1, did the authors from the papers reviewed actually conduct AST on the antibiotics listed?
4. The manuscript should be shortened and focus on the data obtained from NHPs. The sections on the different gene families involved and mechanisms and genetic supports have scientific and nomenclature errors and most of this information is well-known and thus can be omitted from this review.
5. Gene names should be italicized and lowercase, fmrO throughout and beta-lactamase genes, should be written as follows: blaSHV-11. Please correct throughout.
6. Please check the spelling throughout. NHP names are misspelled frequently, as are drug names.
7. The formatting of Table 2 needs revision.
8. Figure 2 needs to be a higher resolution.
Comments on the Quality of English Language
Spelling and English and scientific nomenclature need to be improved significantly.
Author Response
Subject: Revisions
Dear reviewer of Antibiotics Journal,
Thank you for reviewing our work. We are writing this covering letter in accordance with the journal editor's request to explain, point by point, the details of the revisions made to the manuscript and our responses to your comments below. All revisions have been highlighted (in yellow) as suggested.
After receiving your initial comments a few months ago, we have reconsidered certain aspects of our manuscript. You will find in this new version some noticeable changes.
Initially, the scope was expanded to include non-human primates, but we have now narrowed it to focus on African great apes, as our research activities are solely centered on chimpanzees, gorillas, and bonobos. We have also broadened the subject to cover antimicrobial resistance as a whole, rather than focusing solely on resistance genes, which has resulted in an immediate change to the title of our manuscript. Accordingly, we have removed articles dealing with other primates and replaced them with other relevant studies, and we have extended the study period until 2024.
Regarding the sections of the article, we have removed the part dealing with genetic mechanisms and support, as recommended by one of the editors, and added two new sections that compare and discuss:
- Antimicrobial resistance (AMR) between great apes and humans, based on their level of overlap at each site.
- The methodologies used to analyze antimicrobial resistance and their targets.
Comments and Suggestions for Authors
In this review article, the authors evaluated the prevalence of antibiotic resistance genes in non-human primates over a 20 year period. The overall premise of the manuscript is of interest and the data presented are valuable. However, the manuscript needs to be shortened and revised.
1-The estimate is that 10 million people could die from AMR infections by 2050, not 2030, please correct this.
- O'Neill J Tackling drug-resistant infections globally: final report and recommendations. Review on Antimicrobial Resistance, London2016
- O'Neill J Antimicrobial resistance: tackling a crisis for the health and wealth of nations. Review on Antimicrobial Resistance, London2014
Answers-We agree and have corrected
2-Table 1 requires significant attention and reformatting.
"antibiotic resilent" is that supposed to be "antibiotic resistance?"
Answers-No, but the antibiotics tested for sensitivity by the papers's authors and which have developed resistance.
"conjugaison" is that "conjugation"?
"gorilles" should be "gorillas"
gene names should be lower case
beta-lactamase genes should be written as follows blaSHV.
What does "associated mechanism" mean vs. "genetic support"? conjugation and plasmids are way to transfer resistance genes.
Answers-We agree and have corrected.
What does "N" mean? nothing was reported?
Answers: yes!
There are drugs listed in the same sections and resistance mechanisms, and resistance mechanisms listed in the associated mechanism.
Answers-We agree and have corrected.
3- Table 1, did the authors from the papers reviewed actually conduct AST on the antibiotics listed?
Answers-Yes, as described in the Methods of resistance analysis, some authors of the articles reviewed performed a culture-based analysis, resulting in an antibiotic susceptibility test (AST) before proceeding to a specific molecular study using PCR.
4-The manuscript should be shortened and focus on the data obtained from NHPs. The sections on the different gene families involved and mechanisms and genetic supports have scientific and nomenclature errors and most of this information is well-known and thus can be omitted from this review.
Answers- We agree and have removed the part dealing with genetic mechanisms and support have corrected.
- Gene names should be italicized and lowercase, fmrO throughout and beta-lactamase genes, should be written as follows: blaSHV-11. Please correct throughout.
Answers- We agree and have corrected
- Please check the spelling throughout. NHP names are misspelled frequently, as are drug names.
Answers- We agree and have corrected
- The formatting of Table 2 needs revision.
Answers- We agree and have corrected
- Figure 2 needs to be a higher resolution.
8- Answers-We agree and try to reproduce it with a better resolution.
Best regards
Tanguy Tanga.
Reviewer 3 Report
Comments and Suggestions for Authors
The paper “An update on the Genetic characterization of Antibiotic Resistance in Non-Human Primates in Africa” by Tanga et al. could be useful to the scientific community. However, the manuscript is sometimes hard to read and confusing. I had the feeling that it was written in a rush. First, the manuscript is marked with colors (yellow, green, etc.). Second, some Tables are very difficult to read. Moreover, it is unfortunate that the manuscript has no line or page numbers.
Some sentences are not explained. For example, in the Abstract, the sentence
“An overall analysis of AMR between NHPs living in captivity and those living in the wild was not possible due to the nature of the studies, which targeted specific genes.” is unclear. The same idea is inside the paper, and, again, I could not understand why it is impossible to compare the ARG of NHP between those living in the wild and zoos, for example.
The next sentence of the Abstract “The transmission of antibiotic-resistant genes in NHPs is influenced by multiple factors that necessitate thorough investigation through a holistic approach, in order to effectively mitigate their spread”. Does not seem a good conclusion of this work because the authors are not looking for the factors leading to ARG spread…
Materials and Methods.
The sentence “The key words used for the online search were: [antibiotic resistant genes «AND "Mechanisms "AND " Non-Human Primates] " OR [great apes "AND Africa]” is unclear. What is the meaning of “«”? Are you sure that these brackets and the OR are in the correct place?
Consider the sentence: “Only the report of George et al. (2021) explored the entire resistome of the target microbiome. “. The paper George et al. (2021) would find any ARGs present. However, it was not clear whether the other studies would find all ARG possible. For example, if a given study is using PCR to look for BLA genes, the ARGs will not be found. Tanga et al should explain whether the 14 papers would find all ARGs.
The other sentences of Materials and Methods “Due to the specific focus of most of these studies on particular genes and the varying number of individuals tested in each study, it is challenging to conduct an overall analysis of AMR between NHPs living in captivity and those living in the wild. Nevertheless, it has also been reported that animals living in captivity possess a microbiome that is more similar to humans, potentially resulting in a resistome closer to that of humans compared to NHPs living in the wild [15]. Therefore, our analysis primarily focuses on the types of AMR and mechanisms observed in these animals, their origins, ecologies, and their evolution within their ecosystems.” are, again, unclear: why is it challenging to conduct an overall analysis of AMR between NHPs living in captivity and those living in the wild? Moreover, even if that is challenging, why are the authors mentioning this so many times, even in the abstract?
In the first paragraph after the Tables, one can read the sentence “These genes are capable of conducting peptidoglycan synthesis”. Fix this sentence please… because genes encode proteins that do something, not the genes themselves.
In the same page, but four paragraphs below, one can read “The tet genes are Major Facilitator Superfamily (MFS) genes associated with the enzymatic inactivation of tetracycline through antibiotic efflux mechanisms[24].”. Either it is inactivation of tetracycline OR antibiotic efflux mechanisms, I think.
The section “Mechanisms and genetic supports” is about HGT, integrons, plasmids, etc. But then, there are no results. It seems that this text should belong to the Introduction…
Unfortunately, I could not understand whether any of the 14 papers found plasmids encoding for ARGs or integrons with ARGs. This is important.
I liked Fig 2. The idea of those boxes with the genes was very good. But it is difficult to read what is inside those boxes.
About Table2: I think there should be a legend explaining each column head. For example, what do you mean by “% detection gene”? And why do we find always two numbers for each antibiotic?
I think you should rewrite the Conclusions. In my opinion, it is not a conclusion.
Comments on the Quality of English Language
The english seems ok.
Author Response
Subject: Revisions
Dear reviewer of Antibiotics Journal,
Thank you for reviewing our work. We are writing this covering letter in accordance with the journal editor's request to explain, point by point, the details of the revisions made to the manuscript and our responses to your comments below. All revisions have been highlighted (in yellow) as suggested.
After receiving your initial comments a few months ago, we have reconsidered certain aspects of our manuscript. You will find in this new version some noticeable changes.
Initially, the scope was expanded to include non-human primates, but we have now narrowed it to focus on African great apes, as our research activities are solely centered on chimpanzees, gorillas, and bonobos. We have also broadened the subject to cover antimicrobial resistance as a whole, rather than focusing solely on resistance genes, which has resulted in an immediate change to the title of our manuscript. Accordingly, we have removed articles dealing with other primates and replaced them with other relevant studies, and we have extended the study period until 2024.
Regarding the sections of the article, we have removed the part dealing with genetic mechanisms and support, as recommended by one of the editors, and added two new sections that compare and discuss:
- Antimicrobial resistance (AMR) between great apes and humans, based on their level of overlap at each site.
- The methodologies used to analyze antimicrobial resistance and their targets.
Comments and Suggestions for Authors
The paper “An update on the Genetic characterization of Antibiotic Resistance in Non-Human Primates in Africa” by Tanga et al. could be useful to the scientific community. However, the manuscript is sometimes hard to read and confusing. I had the feeling that it was written in a rush. First, the manuscript is marked with colors (yellow, green, etc.). Second, some Tables are very difficult to read. Moreover, it is unfortunate that the manuscript has no line or page numbers.
Some sentences are not explained. For example, in the Abstract, the sentence
1-“An overall analysis of AMR between NHPs living in captivity and those living in the wild was not possible due to the nature of the studies, which targeted specific genes.” is unclear. The same idea is inside the paper, and, again, I could not understand why it is impossible to compare the ARG of NHP between those living in the wild and zoos, for example.
Answers As mentioned in the first part, we have reviewed certain sections of our paper. In the previous version, we described it as being due to the fact that the authors had used different detection methods, ranging from bacterial culture to determine phenotypic antimicrobial resistance, to specific molecular methods for detecting certain genes (PCR), to metagenomics. However, in the new version of our paper, we take into account the comparison/overlap of resistance levels between great apes and humans living very close to or relatively distant from each other. We are also conducting a comparison between captive and wild great apes using the same method, as in the papers by Schaumburg et al.
2-The next sentence of the Abstract “The transmission of antibiotic-resistant genes in NHPs is influenced by multiple factors that necessitate thorough investigation through a holistic approach, in order to effectively mitigate their spread”. Does not seem a good conclusion of this work because the authors are not looking for the factors leading to ARG spread…
Answers - We have reviewed and rewritten our document.
Materials and Methods.
3-The sentence “The key words used for the online search were: [antibiotic resistant genes «AND "Mechanisms "AND " Non-Human Primates] " OR [great apes "AND Africa]” is unclear. What is the meaning of “«”? Are you sure that these brackets and the OR are in the correct place?
Answers -We agree and have corrected.
4-Consider the sentence: “Only the report of George et al. (2021) explored the entire resistome of the target microbiome. “. The paper George et al. (2021) would find any ARGs present. However, it was not clear whether the other studies would find all ARG possible. For example, if a given study is using PCR to look for BLA genes, the ARGs will not be found. Tanga et al should explain whether the 14 papers would find all ARGs.
Answers: We agree and have reviewed/rewritten our document.
5-The other sentences of Materials and Methods “Due to the specific focus of most of these studies on particular genes and the varying number of individuals tested in each study, it is challenging to conduct an overall analysis of AMR between NHPs living in captivity and those living in the wild. Nevertheless, it has also been reported that animals living in captivity possess a microbiome that is more similar to humans, potentially resulting in a resistome closer to that of humans compared to NHPs living in the wild [15]. Therefore, our analysis primarily focuses on the types of AMR and mechanisms observed in these animals, their origins, ecologies, and their evolution within their ecosystems.” are, again, unclear: why is it challenging to conduct an overall analysis of AMR between NHPs living in captivity and those living in the wild? Moreover, even if that is challenging, why are the authors mentioning this so many times, even in the abstract?
Answers: We agree and have reviewed/rewritten our document.
6-In the first paragraph after the Tables, one can read the sentence “These genes are capable of conducting peptidoglycan synthesis”. Fix this sentence please… because genes encode proteins that do something, not the genes themselves.
Answers - We agree and have corrected.
7-In the same page, but four paragraphs below, one can read “The tet genes are Major Facilitator Superfamily (MFS) genes associated with the enzymatic inactivation of tetracycline through antibiotic efflux mechanisms[24].”. Either it is inactivation of tetracycline OR antibiotic efflux mechanisms, I think.
Answers - We agree and have corrected.
8-The section “Mechanisms and genetic supports” is about HGT, integrons, plasmids, etc. But then, there are no results. It seems that this text should belong to the Introduction…
9-Unfortunately, I could not understand whether any of the 14 papers found plasmids encoding for ARGs or integrons with ARGs. This is important.
Answers 8,9- We agree and have reviewed/rewritten our document removing the mechanisms and genetic support’s section.
10-I liked Fig 2. The idea of those boxes with the genes was very good. But it is difficult to read what is inside those boxes.
1 Answers - We agree and have corrected
11-About Table2: I think there should be a legend explaining each column head. For example, what do you mean by “% detection gene”? And why do we find always two numbers for each antibiotic?
Answers-We agree and have corrected. In the previous version, the first numbers were refers to the percentage of a single gene detection (present or absent in the different papers considered) and the second numbers were refers to the percentage of all the genes in the same family. However, in the new version we just keep the family’s percentage.
12-I think you should rewrite the Conclusions. In my opinion, it is not a conclusion.
Answers- We agree and have try to write it better.
Best regards
Tanguy Tanga.
Round 2
Reviewer 2 Report
Comments and Suggestions for Authors
The authors have addressed most of my comments.
Comments on the Quality of English LanguageThere are still some typographical errors.
Author Response
Subject: Revisions
Dear reviewer of Antibiotics Journal,
in accordance with the journal editor's request to explain, point by point, the details of the minor revisions you have highlighted, we are writing this letter again to provide clarification and answer your questions.
Comments and suggestions for authors
comment: There are still some typographical errors.
Answer: We agree and have tried our best to remove the typographical errors and the style of writing.
Best regards
Tanguy Tanga
Reviewer 3 Report
Comments and Suggestions for Authors
The manuscript is now much more focused and clearer.
Some sentences start with numbers, which I think is not appropriate. For example, “10 strains…” or “13 strains…”. I think it should be “Ten strains… “, “Thirteen strains…”, etc,etc
In page 12, one reads “The most detected resistance among AGAs was resistance to beta-lactams, with a detection rate of 36% of targeted studies (Table 2). The most prevalent beta-lactamase genes were blaCTX-M-15 and blaTEM-1 from the bla gene superfamily of beta-lactams, with a detection rate of 28.57%.”.My question is: what is the 28.57%: for both blaCTX-M-15 and blaTEM-1? And 28.57% of all studies, or 28.57% only among beta-lactamase?
Author Response
Subject: Revisions
Dear reviewer of Antibiotics Journal,
in accordance with the journal editor's request to explain, point by point, the details of the minor revisions you have highlighted, we are writing this letter again to provide clarification and answer your questions.
Comments and suggestions for authors
comment-1: The manuscript is now much more focused and clearer. Some sentences start with numbers, which I think is not appropriate. For example, “10 strains…” or “13 strains…”. I think it should be “Ten strains… “, “Thirteen strains…”, etc,etc.
Answer-1: We agree and have corrected.
Comment-2: In page 12, one reads “The most detected resistance among AGAs was resistance to beta-lactams, with a detection rate of 36% of targeted studies (Table 2). The most prevalent beta-lactamase genes were blaCTX-M-15 and blaTEM-1 from the bla gene superfamily of beta-lactams, with a detection rate of 28.57%. ”My question is: what is the 28.57%: for both blaCTX-M-15 and blaTEM-1? And 28.57% of all studies, or 28.57% only among beta-lactamase?
Answer -2: We agree and have corrected. In the previous version of the manuscript, the second digits referred to the percentage of each gene in the same family. We have therefore deleted this, as in the new version we only retain the percentage of the gene family according to all studies.
Best regards
Tanguy Tanga